# Detection of Missing Insulator Caps Based on Machine Learning and Morphological Detection

**DOI:** 10.3390/s23031557

**Published:** 2023-01-31

**Authors:** Zhaoyun Zhang, Hefan Chen, Shihong Huang

**Affiliations:** Electronic Engineering and Intelligence College, Dongguan University of Technology, Dongguan 523000, China

**Keywords:** SVM, missing insulator slices, small-scale dataset, object region detection, machine learning, morphological detection

## Abstract

Missing insulator caps are the key focus of transmission line inspection work. Insulators with a missing cap will experience decreased insulation and mechanical strength and cause transmission line safety accidents. As missing insulator caps often occur in glass and porcelain insulators, this paper proposes a detection method for missing insulator caps in these materials. First, according to the grayscale and color characteristics of these insulators, similar characteristic regions of the insulators are extracted from inspection images, and candidate boxes are generated based on these characteristic regions. Second, the images captured by these boxes are input into the classifier composed of SVM (Support Vector Machine) to identify and locate the insulators. The accuracy, recall and average accuracy of the classifier are all higher than 90%. Finally, this paper proposes a processing method based on the insulator morphology to determine whether an insulator cap is missing. The proposed method can also detect the number of remaining insulators, which can help power supply enterprises to evaluate the degree of insulator damage.

## 1. Introduction

The functions of insulators, which play an important role in high-voltage transmission lines, are to support power lines and provide electrical insulation [1,2,3]. Missing insulator caps are usually found on glass and porcelain insulators, which are mainly caused by tension on insulators from power lines and towers, and erosion of insulators by wind, acid rain and other weather elements. The insulators of these materials age due to accumulated weather erosion. Under the action of power line tension, the insulator cap is damaged and falls off. In addition, glass insulators can eliminate a flashover cap, which is called “zero-value self-explosion”. Thus, the insulator cap will be missing more frequently in glass insulators. A missing insulator cap decreases the mechanical and electrical properties of the whole insulator string, which will threaten power transmission system operations. Therefore, the detection of missing insulator caps is one of the most important topics in transmission line detection.

Computer vision, which is used to detect transmission line images collected by UAV (Unmanned Aerial Vehicles), is the most popular transmission line detection method [4,5,6]. The main reason for its popularity in high-voltage transmission line detection is that its efficiency and safety are higher than those of manual inspection and crewed helicopter detection. Therefore, high-voltage transmission line detection technology based on computer vision has great research significance and practical value [7,8,9].

The method of UAV transmission line detection is shown in Figure 1. It can be divided into two types: detection images in workstation and detection images in UAV [10,11,12]. Transmission line images can be quickly and accurately detected by the detection method of transmission equipment deployed in the workstation. However, this method not only requires the UAV to transmit a large amount of data to the workstation, but struggles to realize the real-time online detection of UAV. In contrast, the detection algorithm deployed in UAV only needs to transmit the information of fault equipment, which can realize the real-time online detection of transmission equipment. However, the computing power of CPU (Central Processing Unit) in UAV is limited, and thus is unsuitable for algorithms that deal with large amounts of data.

Insulator detection methods are mainly based on machine learning methods, morphological detection method and their combination. Machine learning involves designing a learning algorithm to extract experience from samples to optimize computer performance. Morphological detection refers to establishing a model to describe the morphological characteristics of the object to be detected.

Deep-learning-based detection methods can identify and position insulators well. However, due to the small sample size of faulty insulators, deep-learning methods cannot be directly used for insulator fault detection. R.M. Prates [13] designed an image classification method based on a CNN (Convolutional Neural Network) to detect the integrity of four specific types of insulators and used multitask learning to detect the type of insulator and the degree of insulator damage.This method achieved high accuracy in the detection of those four kinds of insulators. However, the proposed method can detect only those four kinds of insulators, which means that it has a slight lack of generalizability. Q. Huang [14] uses artificial intelligence based on a convolution neural network (CNN) to achieve high-precision and high-yield automatic quality detection, highlighting a low-complexity and low-cost neural network architecture based on CNN. However, the CNN architecture is very complex and cannot be implemented on a computing platform with limited resources. J. Zheng [15] proposed a method of transferable feature learning and instance-level adaptation to improve the generalization ability of deep neural networks, which greatly alleviates the challenge of domain transfer between fully labeled source and sparsely labeled target domain points. Q. Zhang [16] used YOLOV3 (You Only Look Once, Version 3) to locate insulator positions and introduced a transfer learning method to train a deep neural network classifier to judge glass insulator self-explosion. To achieve a high recognition rate, SCNS (Stochastic Configuration Networks) and a feedback transfer learning mechanism were introduced into the training of the deep neural network. After adding two kinds of feedback mechanisms, the accuracy rate of this method reached 89%. To solve the problem of detecting missing caps of insulators. X. Tao [17] first used Faster R-CNN (Region-based CNN) to locate a string of insulators and then used another Faster R-CNN to locate the positions of missing caps in insulator images. The disadvantage of this method is that the dataset of a cap cannot meet the training needs of the Faster R-CNN. Therefore, X. Tao proposed a series of methods to change the backgrounds of detection images and expand the dataset several times. In a word, because the normal insulator image is easy to collect, the method based on a deep neural network is usually used to recognize it [18,19]. Due to the limited number of faulty insulator images, if a deep neural network method is adopted, it is necessary to expand the dataset or introduce various mechanisms into the neural network to adapt to small sample detection [20,21].

A deep neural network needs a huge amount of computation, which is usually deployed to edge devices by heterogeneous computing units [22,23,24]. These heterogeneous computing units are still in the research stage in China. Surface-learning-based detection methods do not need to calculate a large number of data operations, so they have more advantages in edge detection than deep learning. X. Wang [25] extracted Gabor features from detection images and trained an SVM classifier to separate insulators from the background. P.S. Prasad [26] combined wavelet transform features and LBP (Local Binary Patterns) features to train an SVM classifier for insulator detection. X. Huang [27] implmented image segmentation of anti-shock hammers using threshold segmentation and morphological processing to find rusted pixels in the segmented images and detected the rust degree of anti-shock hammers by calculating the rust area ratio (RAR) and color difference index (csl). Y. Qiu [28] extracted 935–1725 nm hyperspectral lines and characteristic wavelengths of component spectral lines from hyperspectral images of pollutants and established characteristic spectrum segments of cap and a random forest classification model based on the full-band training data to identify pollution in insulator hyperspectral images. J. Lu [29] designed a classifier based on SVM and realized the detection of a high-voltage line bird nest. The classifier was learned by a multiple SVM ensemble, and the image features of four bird nests were fused. In summary, surface learning classifiers still play an important role in transmission line detection.

In transmission line inspection, the most commonly used machine learning detection methods are RCNN, YOLO and SVM. The detection method based on RCNN has high accuracy, but it needs a long detection time; thus, it is difficult to meet the requirements of real-time detection task. In contrast, YOLO has a Faster detection speed and can carry out real-time detection. However, they have high resource requirements for the deployment environment, and thus are difficult to deploy to the edge devices. Thus, SVM is widely used in edge deployment because its detection performance is better in shallow learners, and it has the advantage of small computing resources.

Morphological processing is one of the most important tools in image processing. The image morphology is a mathematical element with a certain shape, which is used to extract the corresponding shape, texture, color and other information from the image. W. Chen [30] designed an image segmentation algorithm for insulators based on the color characteristics of the insulators, transformed insulator images into binary images and then used the regional pixel threshold method to judge the position of the missing insulator cap. T. Guo [31] proposed a processing algorithm that combines deep learning and morphology. First, Faster R-CNN is used to locate the insulator position, and after the insulator is extracted, the distance between each adjacent insulator cap is measured using the morphological method to determine whether there are any missing insulator caps. W. Pin [32] compared and analyzed traditional machine learning and deep learning image classification algorithms, and found that traditional machine learning is better for solving small sample datasets and the deep learning framework more accurate for recognizing large sample datasets. Y. Zhai [33] analyzed the color features of insulators and proposed a method based on color features to extract aerial insulators from images. According to the arrangement characteristics of insulators, a morphological detection method for missing insulator caps is proposed. It is necessary to segment the insulator string from the image to judge whether there is a missing insulator cap when using the morphological method. Insulator image segmentation techniques mainly include threshold segmentation, cluster segmentation and deep learning image segmentation methods [34,35,36]. Different segmentation methods should be selected according to the needs of insulator fault diagnosis. Clustering segmentation is very effective for cluster classification segmentation and is particularly suitable for insulator segmentation. Although deep-learning-based image segmentation achieves a high segmentation accuracy and good effect, it requires a large training sample. Different segmentation methods should be selected according to the needs of insulator fault diagnosis.

Deep learning detection methods must expand the dataset or introduce various mechanisms in the neural network to accommodate small sample detection when the sample size of faulty insulators is small. Additionally, deep neural networks require a large amount of computation and cannot be deployed directly to edge devices through heterogeneous computing units, so deep learning methods cannot be adapted to small sample insulator fault detection. Therefore, this paper proposes a detection method based on machine learning and morphology to extract grayscale and color features of insulators; extract similar feature regions of insulators from the detection images and generate candidate frames based on these feature regions; and input the captured images into a classifier consisting of SVM (Support Vector Machine) to identify and locate insulators. This method can better quantify the degree of equipment failure and is used in tasks with small sample sizes of faulty insulators to determine if insulator caps are missing, and the proposed method can also detect the number of remaining insulators.

## 2. Proposed Method

The missing insulator cap detection method proposed in this paper is divided into two steps. First, the insulator is located and identified in the transmission line detection image, and then the morphological method is used to determine whether there is a missing insulator cap. The specific steps of this method are shown in Figure 2.

First, similar feature regions of the insulator are extracted from the transmission line inspection image. According to the material characteristics of porcelain and glass insulators and the relevant electrical regulations, it can be concluded that the gray levels of these two insulator images and the values of the red and blue channels in the RGB images are relatively large. Second, these features are used to extract feature regions from the transmission line inspection images. Then, candidate boxes are generated according to these regions, and these boxes are used to separate the candidate images from the inspection images. These candidate images are input into the SVM-based classifier to recognize the insulator, the insulator position is regressed and the target image of the insulator is obtained.

The insulator target image acquisition will be described in detail in Section 3. The structure of the insulator image classifier used in target image acquisition is described in Section 4. Finally, the pixel points belonging to the insulator are segmented from the target image, and the proposed pixel-based statistical method is used to determine the faults. The contents of these steps will be described in detail in Section 5.

The main contributions of this study are as follows. According to the insulator color and gray features, a region search method based on insulator features is proposed. It can remove a large number of insulator-independent regions in the inspection image and simplify the insulator candidate frame extraction. According to the idea of selective search, a regional target detection method is designed. Due to the guidance of insulator feature regions, the insulator candidate feature region can be quickly found, and an insulator target image can be obtained. The K-means clustering algorithm is improved according to the morphological characteristics of the insulator. It is used to remove the background of an insulator target image. Most importantly, a defect detection method for missing insulator caps is proposed. The image data are projected into one-dimensional data that can describe the arrangement of the insulating cap, reducing the amount of calculation needed for defect detection. The detection method in this paper can not only detect the missing position of the insulator cap but can also judge the damage level by calculating the number of remaining insulator caps. In order to prove the robustness of the algorithm, the detection effect of this method under different noises is compared. The fault detection conditions under different shooting angles are analyzed, and the drone adjustment method is provided for the acquisition of insulator keyframes.

## 3. Insulator Target Detection

SVM-based regional positioning usually adopts multiscale windows, edge boxes and BING (binarized normed gradients). However, the background of the inspection image of an insulator is very complicated, so using these methods will slow down the inspection speed. To solve this problem, candidate frame images are generated by extracting similar feature regions of the insulators, which are combined with the SVM classifier to locate the insulators.

### 3.1. Feature Region Extraction

The red porcelain insulator and the blue glass insulator are taken as the research objects, and the feature region extraction method based on color and gray is designed. Insulators made of glass and porcelain have higher grayscale characteristics than the backgrounds. So, the grayscale threshold segmentation can be used to extract their similar feature regions. The grayscale conversion formula of human vision is shown in (2) [37].
(1)$$\mathrm{gray}\left(x,y\right)=0.3*R\left(x,y\right)+0.59*G\left(x,y\right)+0.11*B\left(x,y\right)$$

According to the color features of the two types of insulators, a method based on a color feature map is proposed, as shown in (2).
(2)$${\mathrm{RB}}_{\mathrm{Feat}\left(x,y\right)}=\sqrt{{\left(R\left(x,y\right)-B\left(x,y\right)\right)}^{2}}$$

In (1) and (2), RB_Feat (x, y) represents the value of pixel (x, y) in the RGB color feature image and gray (x, y) represents the value of this pixel in the grayscale image. R(x, y), G(x, y) and B(x, y) represent the points where the position is the (x, y) values of the R, G, and B channels, respectively, of the original image.

The acquisition of the characteristic area is shown in Figure 3. The proposed method is not suitable for high-resolution image processing. Therefore, the subsampled operation is carried out on the insulator inspection image to reduce the image resolution and obtain Figure 3a. Figure 3a is converted into the images shown in Figure 3b,c according to (1) and (2), respectively. Figure 3b,c are segmented using Otsu’s thresholding method to obtain their corresponding binary images, as shown in Figure 3d,e, respectively. The two binary images are subjected to an intersection logic operation, and the combined image is subjected to an open operation of the binary image to obtain a binary representation image of the characteristic image of the insulator, as shown in Figure 3f.

Multiple feature regions can be obtained by detecting the connected regions in the binary image. According to the feature region, candidate images of insulators can be obtained to realize the recognition and location of insulators. The proposed method is not only suitable for red and blue insulator feature region extraction, but also for yellow, cyan, brown and purple insulators. The characteristic of these colors is that the value difference between R channel and B channel is large.

### 3.2. Generation of the Candidate Boxes

According to the binary representation of the insulator features, the original RGB image can be marked with bounding boxes, as shown in Figure 4.

The images captured by these bounding boxes cannot be candidate frames for area object detection because of the following problems:

(1) A string of insulators may be divided into multiple connected regions in the feature map represented by the binary image.

(2) The insulator and the background pixels with similar features to the insulator are in the same communication area.

(3) There is another string of insulators in the candidate box of insulators, leading to missed detections.

To solve these problems, a method similar to a selective search is proposed to merge regions and generate candidate boxes. Although the color and grayscale features of these regions are similar to those of the insulators, they are still very different from the color features of the insulators in the RGB images. Therefore, candidate regions can be determined according to these features. The color feature is judged by the distance from the mean RGB value, and the similarity judgment formula is as follows:(3)$$\mathrm{similarity}=1-\frac{2\sqrt{{\left(R1_{ave}-R2_{ave}\right)}^{2}+{\left(G1_{ave}-G2_{ave}\right)}^{2}+{\left(B1_{ave}-B2_{ave}\right)}^{2}}}{\mathrm{gray}1_{ave}+\mathrm{gray}2_{ave}}$$

This equation represents the similarity judgment of image 1 and image 2. *R*1*_ave_*, *G*1*_ave_* and *B*1*_ave_* represent the average values of the *R*, *G*, and *B* channels in image 1; *R*2*_ave_*, *G*2*_ave_* and *B*2*_ave_* represent the average values corresponding to image 2. According to (3), the candidate frame generation and insulator region location method is as follows:

(1) For each adjacent or intersecting feature box, the similarity is calculated based on the RGB color features of the area connected by the similar features of the insulator in the box. If the similarity exceeds 80%, they will be merged into one box. For two intersecting boxes with the same color feature, the position information of the original box should be retained after the merge to judge whether there are multiple different insulators in a box. This step is repeated until the color feature similarity between each adjacent box is less than this value. (Note: adjacent boxes refer to the distance of the nearest box in each direction and not the nearest box.)

(2) The following process is performed on the image captured by the feature frame generated in Step (1). The image is divided into multiple cells (the central cell in the image and the edge cells on the four sides of the rectangular image, according to the location of the cell). Then, the color characteristics of the image edge cell and the image center cell are compared. If the color feature of each edge cell on one side of the rectangle is less than 40% similar to that of the center cell, all the edge cells on that side are deleted. This step is repeated until one or more cells with the same color characteristics as the center cell appear on the edge.

(3) The images captured by all the boxes are subsampled to a uniform size and then input into the SVM base classifier for classification. All the boxes whose classification results are not insulators are deleted.

(4) For two boxes that intersect and whose captured images are judged as “insulators” by the classifier, we propose a method to judge whether they are different positions of the same string of insulators. The areas of the two intersecting boxes are calculated separately. If the area of a box is less than 20% of the combination of the two boxes, the box is directly regarded as another string of insulators.

## 4. Classifier for Insulator Recognition

The trained SVM base classifier and the candidate frame generation proposed in the previous section can be combined to realize the insulator’s location.

### 4.1. Feature Extraction

Feature engineering refers to the process of transforming the original data into the training data of the model. Image processing can reduce the dimensions of data and obtain better training data features.

#### 4.1.1. LBP

The texture and color feature of insulator string is obviously different from that of the background, so these features can be used for classification and recognition. LBP is an effective texture description operator, and its principle is shown in Figure 5.

First, the original image is converted into a grayscale image, and then the entire image is divided into multiple cells of the same size. Then, the periphery is binarized according to the center of each cell and converted into a decimal number. Finally, by counting the decimal numbers obtained through all the cell operations, the LBP feature histogram of the image is obtained. A binary cyclic shift of the LBP feature data is performed on the histogram, and the smallest decimal number in the displacement is selected to replace the original value to reduce the scope of the histogram. The purpose of this step is not only to preserve the LBP feature rotation data, but also to reduce the amount of input data for the SV.

#### 4.1.2. GLCM

GLCM is a description of image texture. It counts the relationship between two pixels of equal distance in the same direction in the gray image. The acquisition method of GLCM is shown in Figure 6.

It can be seen from this figure that the scale of GLCM is related to the depth of the gray-level image. The gray value range of the original gray image is 0~255, so there are 65,536 kinds of gray relations between two points. The data volume of GLCM can be reduced by a decline in the gray depth of the image, as shown in (4).
(4)$$g=\left\lceil \mathrm{gray}/16\right\rceil$$

The “gray” is the gray image of the original detection, and “g” is the gray image after a decline in the gray depth. In the image g, pixels satisfying a distance of 1 and a direction of 0 degrees are regarded as a group, the relationship between two points in all groups is counted and GLCM is generated.

#### 4.1.3. RGB Histogram

Because the insulator has distinct color features, a RGB histogram is used to extract its color features as the input data of SVM. The RGB histogram is used to calculate the number of pixels of each color depth in the image.

Figure 7 is the RGB histogram of the insulator. The red, blue, and green lines represent the number of pixels for each color depth in the insulator image. It can be clearly seen that most of its pixels have larger values in the red channel, so it is a red insulator. The distribution of 256 groups of color depth in R, G and B channels can be obtained by loading the RGB image of insulator with 8 bit color depth. Therefore, the amount of RGB data input to the classifier is 768.

### 4.2. Structure of Classifier

The classifier for identifying insulators is integrated by four SVM classifiers, and its structure is shown in Figure 8.

The fusion of multiple image features can increase the accuracy of the classifier, but it increases the scale of the classifier at the same time. The proposed classifier integrates SVM classifiers with different image features. It not only improves the accuracy of insulator identification, but also reduces the computational complexity.

SVM is a classifier that performs binary classification on data based on supervised learning. The principle is to find a separating hyperplane that can correctly divide the training dataset and has the largest geometric interval.

The classification principle of the SVM is shown in Figure 9 [38], circles and squares represent different categories, the solid line is the decision boundary between the two categories, and the dotted line represents the interval between the two categories. and the hyperplane is calculated as:(5)w∗x+b=0

*w* is the normal vector of the hyperplane, and *b* is the offset. According to the hyperplane, two decision-making boundaries parallel to it are constructed to realize the two classifications of the samples.
(6)$$\left\{\begin{array}{l} w*x_{i}+b\geq 1\Rightarrow y_{i}=1\\ w*x_{i}+b\leq -1\Rightarrow y_{i}=-1 \end{array}\right.$$

$x_{i}$ represents the i-th sample, and $y_{i}$ represents the classification prediction results of the sample. $\frac{2}{\left\|w\right\|}$ is the distance between the two decision boundaries.

## 5. Insulator Detection

After extracting the feature region of the insulator detection image and using the SVM classifier to judge it, the target insulator image can be taken from the original image. According to the statistical data of the insulator pixel distribution, this method determines whether there is a cap missing in the insulator string. To obtain the insulator pixel statistics, we first use image segmentation technology to convert the target image of the insulator into a binary image.

### 5.1. Segmentation of the Insulator from the Target Image

In the feature region extraction step proposed in this paper, image segmentation technology is used to extract similar feature regions of insulators; however, using this method to segment the target image of insulators will lead to the over-segmentation of insulators. In order to better remove the background of the image and extract the insulator, we propose an improved k-means clustering algorithm to segment the target image for the insulators. The steps of this method are as follows: 

(1) According to the RGB value of the pixel and the position information in the image, all the pixels are transferred to a five-dimensional coordinate system.

(2) Calculate the R + b value of each pixel in the image. The pixel corresponding to the minimum value is randomly selected as the cluster center of cluster 0, and the pixel corresponding to the maximum value is selected as the drama center of cluster 1.

(3) For each pixel, the distance d between it and each cluster center is calculated in the coordinate system. It is divided into a cluster with the nearest cluster center.
(7)$$d=\sqrt{{\left(Rs-Rc\right)}^{2}+{\left(Gs-Gc\right)}^{2}+{\left(Bs-Bc\right)}^{2}+{\left(Xs-Xc\right)}^{2}+{\left(Ys-Yc\right)}^{2}}$$

*Rs*, *Gs* and *Bs* are the *R*, *G* and *B* channel values of the sample pixels, respectively, and *Xs* and *Ys* represent the coordinates of the sample pixels. Similarly, *Rc*, *Gc* and *Bc* are the *R*, *G* and *B* channel values of the cluster center, respectively; *Xc* and *Yc* represent the coordinates of the cluster center.

(4) The mean value of all pixels in each cluster is calculated, and a new cluster center is generated according to the value.

(5) Repeat steps (3) and (4) until the mean value no longer changes or the range of change is less than the threshold value. Finally, according to the grouping information of each pixel, a segmented binary image of the insulator is generated.

Since the method knows the color characteristics of the insulator, it can choose the points with the largest and smallest sums of the R and B channel components as the two initial cluster centers in the k-means clustering algorithm. In this way, the problem of slow fitting due to the random selection of initial points can be avoided.

Figure 10a,c,e show the most common types of insulators made from glass and porcelain; the missing cap phenomenon often occurs in these three types of insulators. The binary images obtained after the k-means clustering segmentation of Figure 10a,c,e are shown in Figure 10b,d,f, respectively. Although there is still some noise and over segmentation in the segmented binary image, these factors do not affect the “missing insulator-cap” judgment.

### 5.2. Determine Whether the Insulator Cap Is Missing

Analyzing the distribution characteristics of the insulator pixels in a binary image of the insulator can determine whether a cap is missing. In this paper, a new insulator pixel distribution method is proposed. According to the shape characteristics of the insulator, the extracted insulator image is first rotated so that the angle between the central axis of the insulator strip and the horizontal axis of the image is less than 30°. Second, the number the insulator pixels in each column of the insulator binary image is counted. Because the insulator pixels are given values of one and the background pixels are given values of zero after the image is segmented, the statistical formula for each column of insulator pixels is as follows:(8)$$yi=\sum_{j=0}^{N-1}A[i][j]$$

In (8), M * N is the resolution of the binary image, and *A* [i] [j] is the value of the pixel in the i-th row and j-th column. The binary image and its corresponding insulator pixel content statistics in each column are shown in Figure 11. It can be seen from the figure that when the insulator segmentation effect is ideal, the missing position of the cap can be quickly found from the pixel statistics chart.

In the pixel statistical diagram of an insulator, the approximate position of the abscissa of the missing cap can be quickly located. The position in the RGB image of the insulator missing cap is shown in the red box in Figure 11f. In Figure 11b, the distance between adjacent peaks is almost equal, and the wave crest of the missing cap position is obviously much smaller than those of the other peaks. Since the insulators in the figure are arranged in double strings, it is necessary to return to the binary image to locate the “missing insulator-cap”. The abscissa range of the missing cap is obtained according to the pixel statistical graph, and then the ordinate of the missing cap is further determined according to the arrangement of insulator caps and two strings of insulators in the abscissa range. Due to the influence of complex background factors, there may be some noise in the segmented binary image, as shown in Figure 11d. However, the location of the missing cap can still be clearly found in its pixel content statistical graph, as shown in Figure 11e.

Through the statistical graph of insulator pixels, the phenomenon of the missing cap can be judged. However, for insulators installed in parallel, the number of remaining insulator caps cannot be calculated. Therefore, it is necessary to further process the binary image of the insulator.

For example, in a high-voltage line, M insulators are installed in parallel, and each insulator has N insulator caps. According to the statistical graph of insulator pixels, the binary image of insulator is cut into N images. The connected regions in the segmented image whose area reaches the set threshold are marked, and the center coordinates of all marked connected regions are calculated. Due to the camera angle and image segmentation, a connected region may contain multiple insulator caps. Therefore, it is necessary to calculate the area of connecting area to judge how many insulator caps this area represents. By comparing the number and area of connected regions and the position of connected regions in n images, the position of missing caps and the number of remaining caps can be obtained.

## 6. Experimental Section

### 6.1. Environment and Dataset 

The experimental operating system is ubuntu, and the deployment code of the proposed insulator detection method is python 3.5. The classifier for insulator recognition is constructed using scikit-learn and OpenCV is used to extract image features.

We collect 500 insulator detection images and use the insulator feature area extraction method to generate candidate boxes. Among these boxes, we manually select the box belonging to the insulator and the background box. Finally, we manually label the images captured by these boxes and generate a dataset. The dataset is divided into two categories: “insulators” and “background”. The “background” is mainly the background target detected in the detection image through the proposed similar feature area of the insulator, and there are a total of 450 background images. The “insulators” include 225 porcelain insulators and 225 glass insulators. When training the SVM classifier, the insulator and background samples are randomly divided into a training set and test set at a ratio of 7:3. The required image features are extracted and the four basic classifiers of the ensemble classifier are trained, respectively.

### 6.2. Penalty Parameter “C”

To solve the overfitting problem of the SVM and improve the generalization ability of the classifier, the penalty function “*C*” is introduced to relax the edges, and the samples with incorrect classifications are treated as noise. The classification deviation of the sample by the SVM classifier can be expressed as a loss function:(9)loss=max(0,1−y(wx+b))

After introducing the relaxation variable, the optimization problem mentioned above can be transformed into the following formula:(10)$$\min_{w,b,\zeta }=\frac{1}{2}{\left\|w\right\|}^{2}+C\sum_{i=1}^{n}\zeta i$$

The formula follows the following rule:(11)$$y_{i}(wx_{i}+b)\geq 1-\zeta_{i}(\zeta_{i}\geq 0)$$

*C* is the penalty factor and *w* is the hyperplane in (11). 

$\zeta_{i}$ is the classification loss of the i-th sample and $\sum_{i=1}^{n}\zeta_{i}$ is the composite error. The value of *C* represents the attention given to the sample value during training. When *C* is infinite, samples with classification errors are not allowed, so this problem becomes a hard-boundary SVM problem. When *C* infinitely approaches 0, the formula no longer focuses on whether the classification of the sample is correct, and the SVM loses the meaning of the classification. The debugging process of the penalty factor *C* is shown in Figure 12.

The basic classifier of the proposed classification framework is independent training. All the basic classifiers are trained and debugged to the best performance before ensemble detection. In the proposed classification framework, the penalty parameters of four basic learners are 2, 4, 16 and 32, respectively.

### 6.3. Comparison with a Single-Image Feature Classifier

The common Surface classifiers are SVM, KNN (K-Nearest Neighbor) and RF (Random Forest). The comparison results between the detection method proposed in this paper and common machine learning classifier methods are shown in Table 1.

It can be seen from the table that the accuracy, recall and AP (Average Precision) of the classifier integrated with multiple features are higher than those of the unary classifier with only one input feature.

### 6.4. Compared with Ensemble Classifier or Multi-Feature Classifier

The comparative experiments of multi-feature fusion machine learning detection and multi-classifier ensemble detection are shown in Table 2.

The LBP input into the classifier integrated by SVM, RF and KNN has higher accuracy than that of the SVM classifier based on LBP. This shows that the performance of the ensemble classifier is better than that of the unit classifier. For multi-feature fusion classifier, it is necessary to evaluate the performance of all features. Combining the image features that are not suitable for insulator classification will lead to a decline in classification accuracy. The proposed classification framework in this paper integrates three features, and its classification performance is similar to that of a single SVM classifier based on the fusion of these features. In contrast, the proposed classifier is faster and employs fewer computing resources. This is conducive to the deployment of edge computing.

### 6.5. Verification of Our Detection Method

To verify the feasibility of the approach, the proposed method detects 100 insulator inspection images, half of which are missing cap insulators. The detection results are shown in Table 3.

The detection accuracy is 90%, which meets the detection requirements of transmission lines in China. The main reason for the failure of detection is the lack of insulator target detection due to the low accuracy of the classifier. For the insulator detected by the proposed target detection method, we can almost accurately detect whether it is missing a cap. The insulators tested by the proposed method are shown in Figure 13. Only the common insulator test in Figure 13a was missed, and all other pictures were tested correctly.

In the detection work, the drone shakes due to the effect of the airflow, which affects the detection image. In order to verify the robustness of the proposed detection method, 100 pictures were processed with noise before fault detection. The detection results are shown in Table 4.

It can be seen from the table that the noise has a certain influence on the detection accuracy of insulators. However, the detection accuracy is still higher than 80%, so the proposed method has a certain anti-interference ability.

## 7. Discussion

### 7.1. Key Frame Acquisition

The proposed detection method can quickly and accurately locate a missing insulator cap in an insulator string when the distance between each insulator cap is similar. Although the specifications of transmission line insulators require that the distance between each insulator cap is equal, the distance between insulators will be considerably different due to the different shooting angles of patrolling UAVs or line-walking robots. In addition, insulators photographed from certain angles have the problem of blind spots in their field of view. In theory, it is impossible to detect defects through image processing. In the actual insulator detection work, the “key frame” acquisition technology is used to solve this problem.

Figure 14 shows the insulator detection images taken by the camera at different angles. The spacing between the insulator caps in Figure 14a is clear, and this can be used as a key frame for defect detection. In Figure 14b, the cap of the insulator is partially blocked, and the spacing is very small. It can only detect the defect position and cannot detect the number of remaining caps. In Figure 14c, one row of insulators is completely shielded by the other row of insulators, and defect detection cannot be performed.

Since the position of the insulator is known, the best shooting position of the insulator detection image can be determined according to the type of tower. Therefore, most of the inspection images are of the type shown in Figure 14a,b. In order to obtain better detection results, a method for judging camera angles is proposed. This method can provide the UAV with adjusted direction navigation through image processing technology.

The adjustment method of the shooting position is shown in Figure 15. The camera position of the insulator in Figure 15a is to the right, so the distance between the insulator cap on the left of the insulator is small, which makes it impossible to judge the position between each cap from Figure 15c. Therefore, the offset between the camera and the key frame position can be derived from the pixel statistics of the insulator. At this time, the drone moves to the left until the distance between the insulator caps is equal. Therefore, the shooting position of the drone can be automatically adjusted according to the information obtained via the insulator pixel statistical map.

### 7.2. Degree of Damage of the Insulator

The replacement of insulators in transmission lines requires that power is off, so the scheduling of insulator replacement work is very important. Compared with the positioning of the missing insulator cap, the power supply company is more concerned about whether the insulation capacity of the insulator is still qualified. The number of insulator cap is specified based on the different voltages of transmission lines. In China, the number of glass insulators for 110 kV transmission lines cannot be less than 7, and the number of insulators in 500 kV transmission lines cannot be less than 25. Moreover, the number of insulator caps made of different materials is different under the same voltage. The power supply company deploys insulators according to these regulations, and the number of insulator cap deployed is usually greater than the specified number. When the number of insulator caps is lower than that specified, the power company must replace the insulators.

The proposed detection method is capable not only of locating the position of the missing cap, but also of calculating the number of remaining caps. In the previous research on missing insulator caps, only the position of missing insulator caps was detected, and the detection of the remaining insulator cap was ignored. According to the number of remaining insulators to be detected and the voltage of the transmission line, the fault degree of insulators can be classified. This method is helpful for power supply enterprises, allowing them to create a replacement plan according to the degree of insulator damage.

## 8. Conclusions

According to the gray and color features of insulators, this paper proposes a method for insulator feature region extraction and candidate frame generation. Then, the method inputs the pictures captured by these boxes into the SVM base classifier to identify the insulators. Finally, a method for the subregional judgment of insulator pixels is proposed to determine whether there is a missing insulator cap and to locate the position of any missing caps in the string of insulators. The proposed feature region extraction method can effectively extract feature regions of insulators and generate candidate frames. The accuracy, average accuracy and recall rate of the proposed classifier are higher than 90%, which meets the performance requirements of power supply enterprises in China. The accuracy of this method is 90%, and most of the errors are caused by omission detect of insulators.

Aiming at the robustness of the algorithm in practical applications, a key frame extraction scheme in the image acquisition process is proposed. Developers can write control algorithms for UAVs to automatically obtain key frames based on the proposed method for determining the shooting angle of insulators. In addition, the number of remaining insulator caps can be detected after obtaining the pixel statistics chart of the insulators. Compared with other insulator detection methods, this method is more suitable for the actual detection needs of high voltage lines.

## Figures and Tables

**Figure 1 sensors-23-01557-f001:**
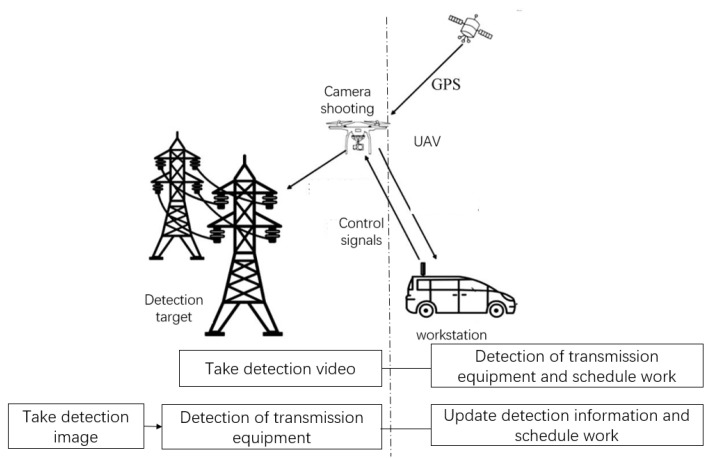
UAV inspection of transmission line.

**Figure 2 sensors-23-01557-f002:**
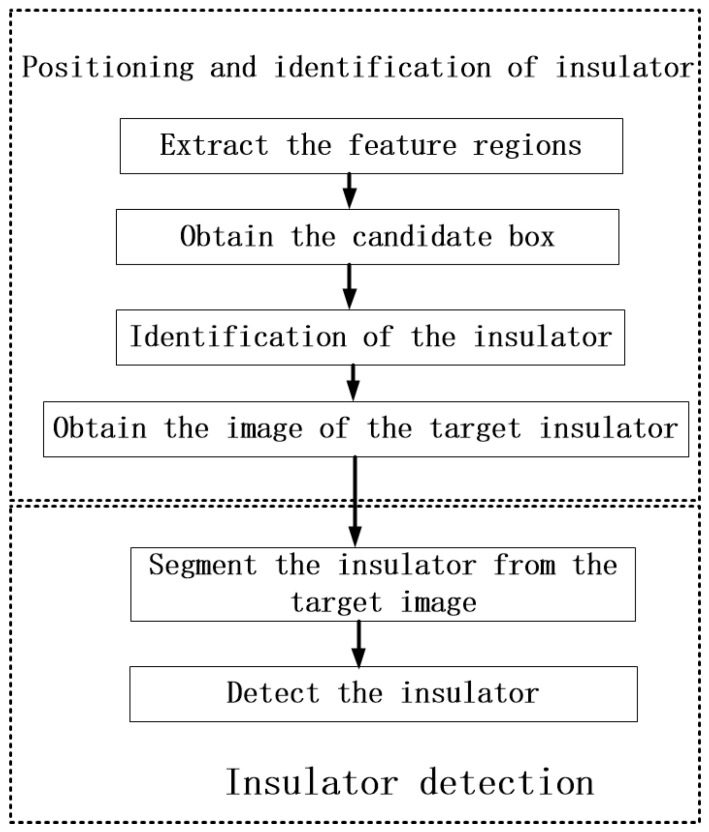
Insulator detection method.

**Figure 3 sensors-23-01557-f003:**
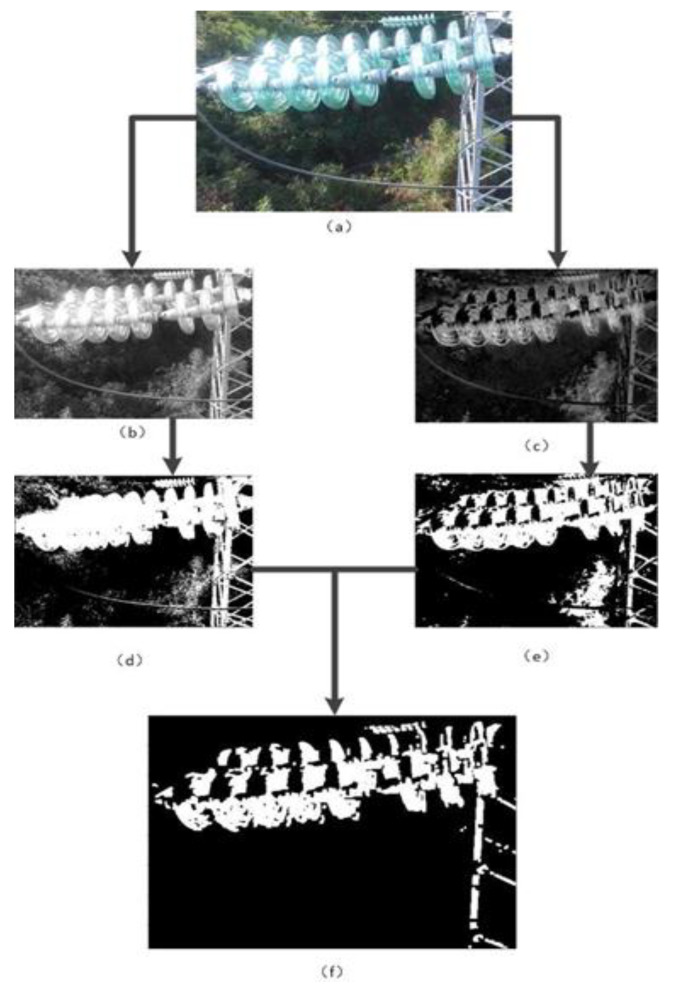
The process of obtaining similar areas of insulators. (**a**) Reduced insulator inspection image, (**b**) grayscale insulator detection image, (**c**) “RB color” feature insulator detection image, (**d**) binary image for gray-level threshold segmentation, (**e**) binary image for “RB color” feature image threshold segmentation, and (**f**) binary image representation of the insulator feature image.

**Figure 4 sensors-23-01557-f004:**
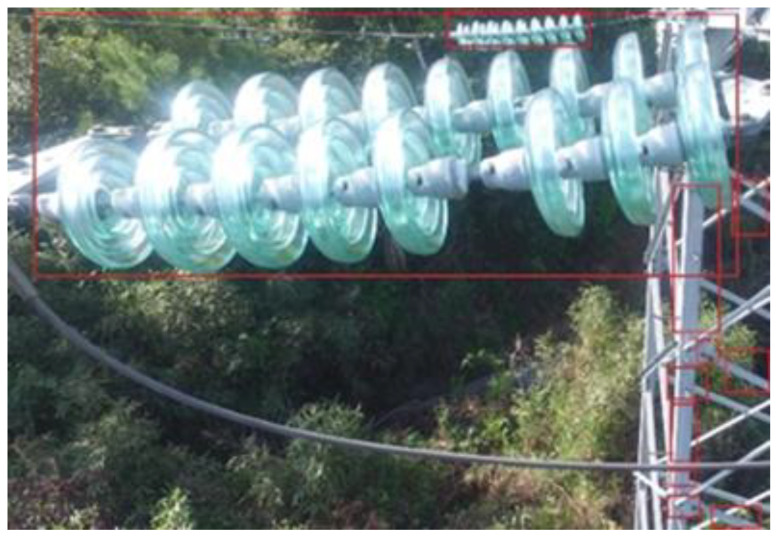
Generation of the minimum bounding box of the feature area.

**Figure 5 sensors-23-01557-f005:**
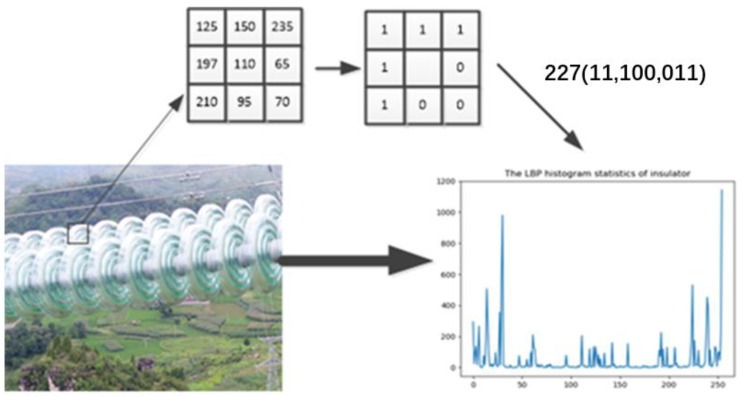
LBP feature histogram extraction.

**Figure 6 sensors-23-01557-f006:**
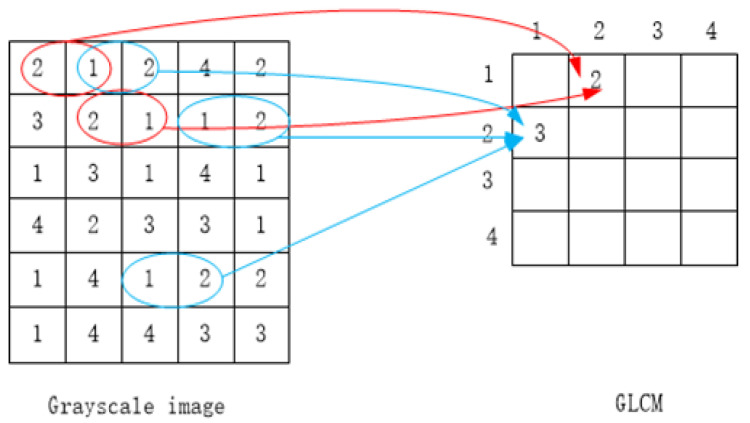
Acquisition of GLCM.

**Figure 7 sensors-23-01557-f007:**
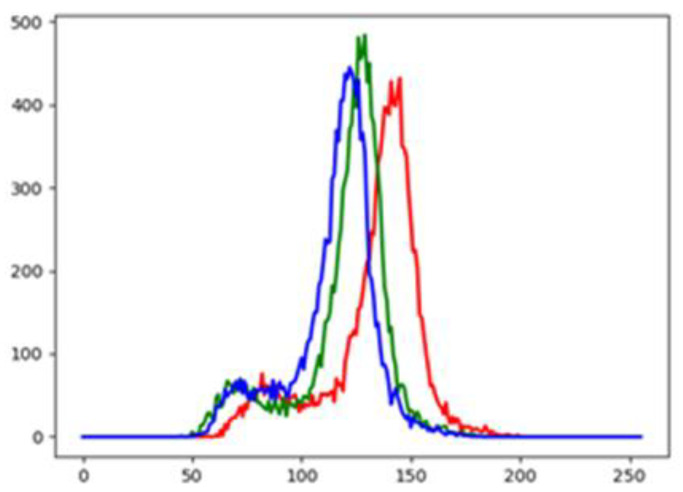
RGB histogram of insulator.

**Figure 8 sensors-23-01557-f008:**
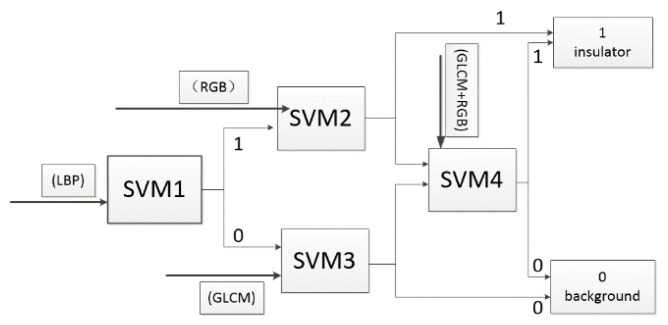
The structure of the classifier.

**Figure 9 sensors-23-01557-f009:**
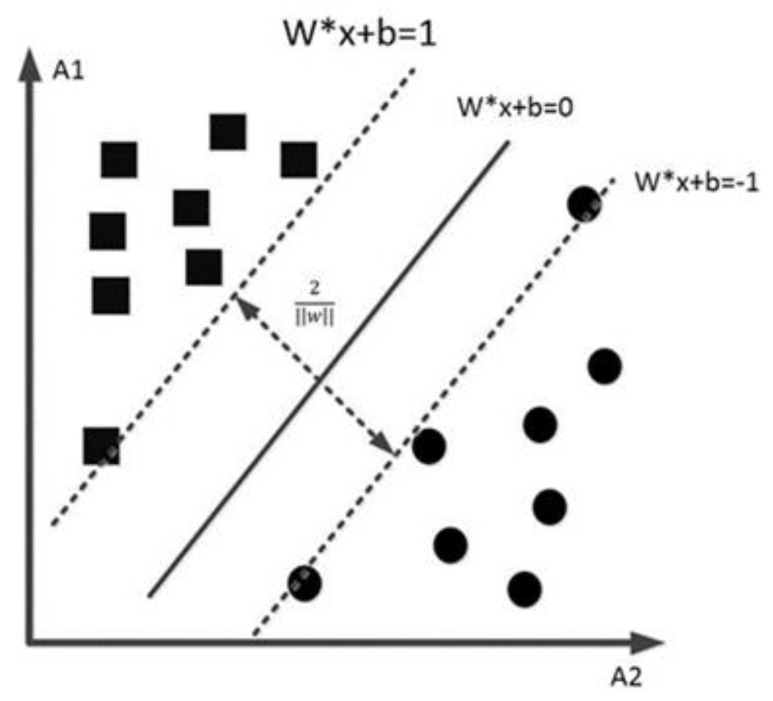
Principle of the SVM classifier.

**Figure 10 sensors-23-01557-f010:**
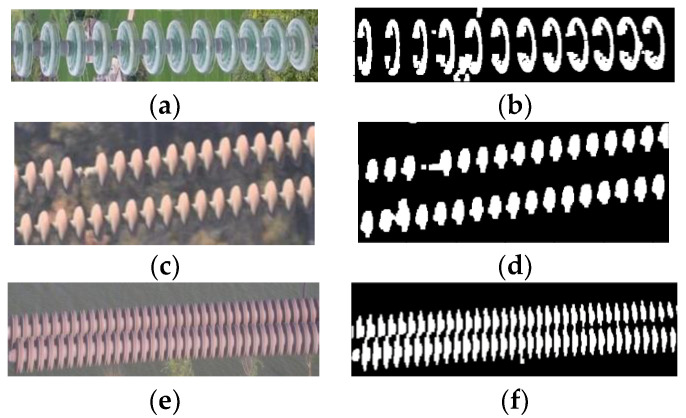
Segmentation effects of different types of insulators. (**a**) Glass insulators, (**b**) binary image of glass insulators, (**c**) porcelain insulators 1, (**d**) binary image of porcelain insulators 1, (**e**) porcelain insulators 2, (**f**) binary image of porcelain insulators 2.

**Figure 11 sensors-23-01557-f011:**
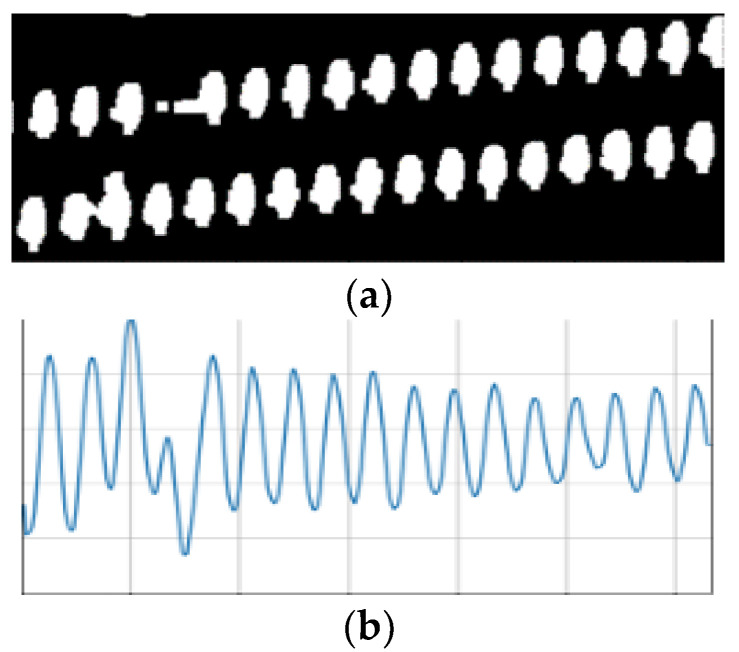
Binary images of insulators and their corresponding pixel statistical charts. (**a**) Binary image of insulators, (**b**) pixel statistical chart, (**c**) location of the missing cap in the RGB image, (**d**) binary image of insulators with noise, (**e**) pixel statistical chart with noise, (**f**) location of the missing cap in the RGB image.

**Figure 12 sensors-23-01557-f012:**
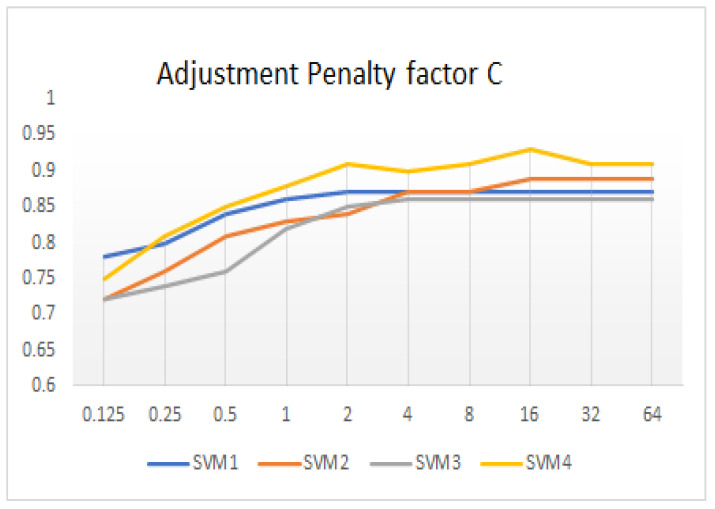
Penalty factor C debugging.

**Figure 13 sensors-23-01557-f013:**
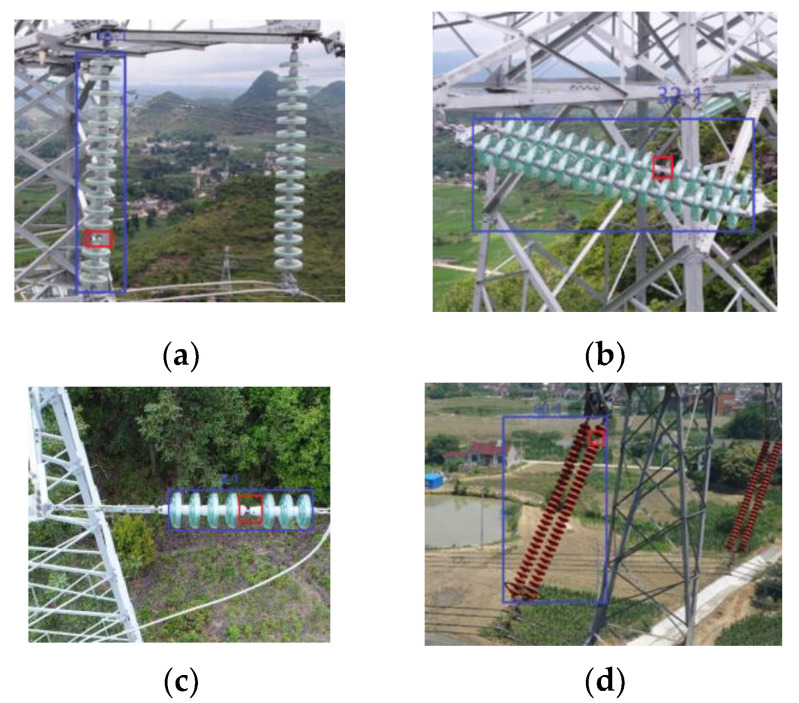
Missing insulator cap detection. (**a**) Glass insulator with 1 * 16 installation, (**b**) glass insulator with 2 * 16 installation, (**c**) glass insulator with 1 * 8 installation, (**d**) porcelain insulator with 2 * 30 installation.

**Figure 14 sensors-23-01557-f014:**
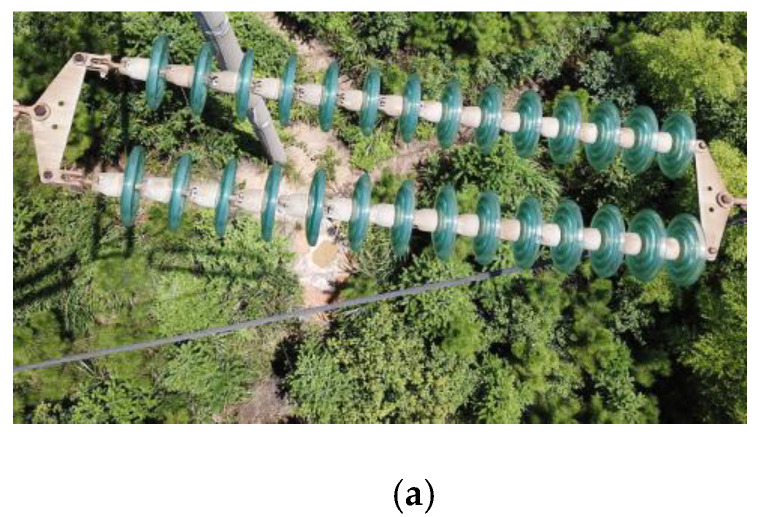
Insulator detection at different angles. (**a**) Two insulators without overlapping parts, (**b**) Two insulators with partial overlap, (**c**) Two insulators with complete overlap.

**Figure 15 sensors-23-01557-f015:**
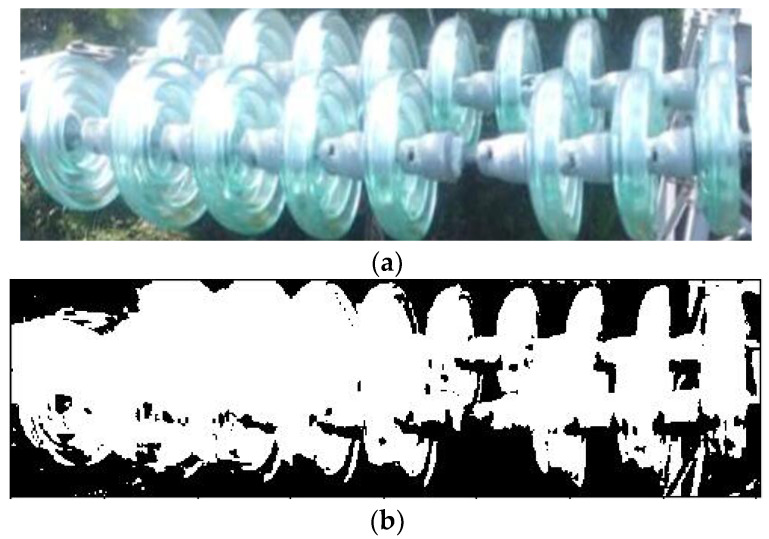
Influence of shooting angle. (**a**) Insulator with the shooting angle to the right, (**b**) binary graph with unsatisfactory insulator segmentation effect, (**c**) pixel statistical chart of the insulator (the segmentation effect is not ideal).

**Table 1 sensors-23-01557-t001:** Compared with single image feature classifier.

Classifier	Accuracy	AP	Recall
Our method	0.9333	0.9076	0.9433
LBP-SVM	0.8716	0.8040	0.8867
RGB-SVM	0.8938	0.8394	0.8867
GLCM-SVM	0.8628	0.7926	0.8773
HOG-SVM	0.7155	0.6624	0.6460
LBP-RF	0.8672	0.7024	0.8773
RGB-RF	0.8672	0.7830	0.7452
GLCM-RF	0.8539	0.7779	0.8867
HOG-RF	0.7244	0.6544	0.6857
LBP-KNN	0.8903	0.8475	0.8584
RGB-KNN	0.7831	0.7830	0.7452
GLCM-KNN	0.8716	0.8040	0.8867

**Table 2 sensors-23-01557-t002:** Compared with ensemble classifier or multi feature classifier.

Classifier	Accuracy	AP
LBP →SVM + RF + KNN (voting)	0.90	0.85
LBP + RGB→SVM	0.91	0.85
LBP + RGB + GLCM →SVM	0.94	0.88
LBP-SVM + RGB-SVM + GLCM-SVM (voting)	0.93	0.86
LBP + HOG → SVM	0.85	0.80
LBP + RGB + GLCM + HOG→ SVM	0.91	0.86
Our method	0.93	0.90

**Table 3 sensors-23-01557-t003:** Compared with ensemble classifier or multi feature classifier.

Test Object	Total	Omission	False	Accuracy
missing-cap insulator	50	5	1	0.90
Normal insulator	50	4	0	

**Table 4 sensors-23-01557-t004:** The influence of noise on the detection effect.

Noise	Variance	Omission	False	Accuracy
Gaussian	0.001	9	1	0.90
Gaussian	0.005	9	1	0.88
Gaussian	0.01	12	3	0.85
Poisson	/	10	1	0.89
Salt and pepper	0.001	9	1	0.90
Salt and pepper	0.005	11	2	0.87
Salt and pepper	0.01	12	8	0.80

## Data Availability

The data that support the findings of this study are available from the corresponding author upon reasonable request.

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
