# Peer review of "Detection of Missing Insulator Caps Based on Machine Learning and Morphological Detection"

_sensors, 2023, doi:10.3390/s23031557_

Round 1

Reviewer 1 Report

1. similar works should be cited and compared with this work, such as

[1] Medak D, Milković F, Posilović L, et al. Detection of Defective Bolts from Rotational Ultrasonic Scans Using Convolutional Neural Networks[C]//2022 27th International Conference on Automation and Computing (ICAC). IEEE, 2022: 1-6.

[2] Huang Q. High-precision quality inspection for screws using artificial intelligence technology[J]. Proceedings of the CIB W78-International Council for Research and Innovation in Building and Construction, Newcastle, UK, 2019: 18-20.

2. Do you use transfer learning techniques to improve the generalibity of your neural network, such as 

[1] Zheng J, Lu C, Hao C, et al. Improving the generalization ability of deep neural networks for cross-domain visual recognition[J]. IEEE Transactions on Cognitive and Developmental Systems, 2020, 13(3): 607-620.

[2] Hao C, Chen D. Software/Hardware Co-design for Multi-modal Multi-task Learning in Autonomous Systems[C]//2021 IEEE 3rd International Conference on Artificial Intelligence Circuits and Systems (AICAS). IEEE, 2021: 1-5.

3. Do you consider using weight quantization or network pruning techniques to reduce the complexity and memory usage of your network?

Such as 

[1] https://www.tensorflow.org/lite/performance/post_training_quantization

[2] Tang Z, Luo L, Xie B, et al. Automatic Sparse Connectivity Learning for Neural Networks[J]. IEEE Transactions on Neural Networks and Learning Systems, 2022.

4. What techniques or ideas do you use to reduce the computational complexity or memory usage of your neural networks?   5. What is the technical contribution of your work? How many percents of trainable parameters do you reduce? How faster can you run your model for inference?

Author Response

Thank you for your kind review. We revised the manuscript in accordance with yours comments, and carefully proof-read the manuscript to minimize typographical, grammatical, and bibliographical errors.

Reviewer 2 Report

In this study, a detection method for insulator missing-cap based on machine learning (SVM algorithm) and morphological detection is proposed. The proposed method is compared with many common machine learning classifier methods and several ensemble classifier or multi feature classifier. Results shows that the proposed method performs best. Several minor questions are as follows:

1.     The comparisons with many common machine learning classifier methods and several ensemble classifier or multi feature classifier show that the proposed method is the best.

a)       Please explain in theory the reason why the proposed method performs so well, so as to improve the theoretical value of the study.

b)      Please give the references of the methods for comparisons in Table 1 and Table 2.

c)       What programming language and software are used to realize the methods for comparisons in Table 1 and Table 2? Can the code be shared with the paper?

2.     What is the relationship between the listed studies in Introduction Section and the proposed methods in Proposed method Section? Or what problem has the proposed method solved in the existing research? Please explain it clearly in Introduction Section to highlight the innovation of this study.

3.     Some pictures in the text are too blurred, please update, such as Figure 1, and so on.

4.     There are many text errors. Please check and correct them in full text carefully. For example:

a)       The sentence in Lines 51 and 53 was repeated twice.

b)      There is one more letter at the beginning of Line 133.

c)       The first letter is missing in Line 614, and there is one more full stop in Line 615.

d)      Not listed one by one.

Author Response

(The authors gave the same response as above.)

Round 2

Reviewer 2 Report

I do not have any more question.